

**Open burning of rice, corn and wheat straws: primary**
**emissions, photochemical aging, and secondary organic aerosol**
**formation**
Zheng Fang[1,3], Wei Deng[1,3], Yanli Zhang[1,2], Xiang Ding[1], Mingjin Tang[1], Tengyu Liu[1], Qihou
Hu[1], Ming Zhu[1,3], Zhaoyi Wang[1,3], Weiqiang Yang[1,3], Zhonghui Huang[1,3], Wei Song[1,2], Xinhui
Bi[1], Jianmin Chen[4], Yele Sun[5], Christian George[6], Xinming Wang[1,2,*]
[1]State Key Laboratory of Organic Geochemistry and Guangdong Key Laboratory of
Environment Protection and Resources Utilization, Guangzhou Institute of Geochemistry,
Chinese Academy of Sciences, Guangzhou 510640, China
[2]Center for Excellence in Regional Atmospheric Environment, Institute of Urban Environment,
Chinese Academy of Sciences, Xiamen 361021, China
[3]University of Chinese Academy of Sciences, Beijing 100049, China
[4]Shanghai Key Laboratory of Atmospheric Particle Pollution and Prevention, Department of
Environmental Science & Engineering, Fudan University, Shanghai 200433, China
[5]Institute of Atmospheric Physics, Chinese Academy of Sciences, Beijing 100029, China
[6]Institut de Recherches sur la Catalyse et l'Environment de Lyon (IRCELYON), CNRS,
UMR5256, Villeurbanne F-69626, France
*Correspondence to: X. Wang (wangxm@gig.ac.cn)*



**Abstract.** Agricultural residues are among the most abundant biomass burned globally,
especially in China. However, there is rare information on primary emissions and
photochemical evolution of agricultural residues burning. In this study, indoor chamber
experiments were conducted to investigate primary emissions from open burning of rice, corn
and wheat straws and their photochemical aging as well. Emission factors of $NO_x$, $NH_3$, $SO_2$,
non-methane hydrocarbons (NMHCs), particulate matter (PM), organic aerosol (OA) and
black carbon (BC) under ambient dilution conditions were determined. Olefins accounted
for >50% of the total NMHCs emission (2.47 to 5.04 g kg$^{-1}$), indicating high ozone formation
potential of straw burning emissions. Emission factors of PM (3.73 to 6.36 g kg$^{-1}$) and primary
organic carbon (POC, 2.05 to 4.11 gC kg$^{-1}$), measured at dilution ratios of 1300 to 4000, were
lower than those reported in previous studies at low dilution ratios, probably due to the
evaporation of semi-volatile organic compounds under high dilution conditions. After
photochemical aging with OH exposure range of $(1.97\text{-}4.97)\times10^{10}$ molecule cm$^{-3}$ s in the
chamber, large amounts of secondary organic aerosol (SOA) were produced with OA mass
enhancement ratios (the mass ratio of total OA to primary OA) of 2.4-7.6. The 20 known
precursors could only explain 5.0-27.3% of the observed SOA mass, suggesting that the major
precursors of SOA formed from open straw burning remain unidentified. Aerosol mass
spectrometry (AMS) signaled that the aged OA contained less hydrocarbons but more oxygen-
and nitrogen-containing compounds than primary OA, and carbon oxidation state ($OS_c$)
calculated with AMS resolved O/C and H/C ratios increased linearly ($p<0.001$) with OH
exposure with quite similar slopes.



## 1 Introduction

On the global scale, biomass burning (BB) is the main source of primary organic carbon (OC)
(Bond et al., 2004; Huang et al., 2015), black carbon (BC) (Cheng et al., 2016), and brown
carbon (BrC) (Laskin et al., 2015). It is also the second largest source of non-methane organic
gases (NMOGs) in the atmosphere (Yokelson et al., 2008; Stockwell et al., 2014). In addition,
atmospheric aging of biomass burning plumes produces substantial secondary pollutants. The
increase of tropospheric ozone ($O_3$) in aged biomass burning plumes could last for days and
even months (Thompson et al., 2001; Duncan et al., 2003; Real et al., 2007) with complex
atmospheric chemistry (Arnold et al., 2015; Müller et al., 2016). Moreover, biomass and
biofuel burning could contribute up to 70% of global secondary organic aerosols (SOA) burden
(Shrivastava et al., 2015) and hence influence the seasonal variation of global SOA (Tsigaridis
et al., 2014). Since it produces large amounts of primary and secondary pollutants, it is essential
to characterize primary emissions and photochemical evolution of biomass burning in order to
better understand its impacts on air quality (Huang et al., 2014), human health (Alves et al.,
2015) and climate change (Andreae et al., 2004; Koren et al., 2004; Laskin et al., 2015; Huang
et al., 2016).

60        Open burning of agricultural residues, a convenient and inexpensive way to prepare for

the next crop planting, could induce severe regional haze events (Cheng et al., 2013; Tariq et
al., 2016). Among all the biomass burning types, agricultural residues burning in the field is
estimated to contribute ~10% of the total mass burned globally (Andreae and Merlet, 2001),
and its relative contribution is even larger in Asia (~34%) and especially in China (>60%)
(Streets et al., 2003) where >600 million people live in the countryside (NBSPRC, 2015).
Agricultural residues burned in China were estimated to be up to 160 million tons in 2012,
accounting for ~40% of the global agricultural residues burned (Li et al., 2016). As estimated
by Tian et al. (2011), agricultural residues burning contributed to 70-80% of non-methane



hydrocarbons (NMHCs) and particulate matter (PM) emitted by biomass burning in China
during 2000-2007. A better understanding of the role agricultural residual burning plays in air
pollution in China and elsewhere requires better characterization of primary emission and
atmospheric aging of emitted trace gases and particles for different types of agricultural
residues under different burning conditions.

In the pasted two decades, there have been increasing numbers of characterization of

biomass burning emissions. Andreae and Merlet (2001) summarized emission factors (EFs) for
both gaseous and particulate compounds from seven types of biomass burning. Akagi et al.
(2011) updated the emission data for fourteen types of biomass burning, and newly identified
species were included. Since biomass types and combustion conditions may differ in different
studies, reported emission factors are highly variable, especially for agricultural residues
burning (Li et al., 2007; Cao et al., 2008; Zhang et al., 2008; Li et al., 2009; Yokelson et al.,
2011; Brassard et al., 2014; Sanchis et al., 2014; Wang et al., 2014; Ni et al., 2015; Kim Oanh
et al., 2015; Li et al., 2017). Moreover, previous studies on agricultural residues burning were
mostly carried out near fire spots or in chambers with low dilution ratios. Since biomass
burning organic aerosols (BBOA) are typically semi-volatile (Grieshop et al., 2009b; May et
al., 2013), it is expected that measured BBOA emission factors would be affected by dilution
processes (Lipsky et al., 2006), and BBOA emission factors under ambient dilution conditions
are still unclear. Furthermore, knowledge in NMOGs emitted from agricultural residues
burning is very limited. As reported by Stockwell et al. (2015), ~21% (in weight) of NMOGs
in biomass burning plumes have not been identified yet. Therefore, comprehensive
measurement and characterization of gaseous and particulate species emitted by agricultural
residues burning under ambient dilution conditions are urgently needed.

Great attention has been drawn to SOA formation and transformation in biomass burning

plumes recently, since significant increase of mass and apparent change in physicochemical





characteristics of aerosols have been observed during atmospheric aging of biomass burning
plumes in both field and laboratory studies (Grieshop et al., 2009a,b; Hennigan et al., 2011;
Heringa et al., 2011; Lambe et al., 2011; Jolleys et al., 2012; Giordano et al., 2013; Martin et
al., 2013; Ortega et al., 2013; Ding et al., 2016a; Ding et al., 2016b; Ding et al., 2017). For
agricultural residues burning, evolution processes have not been well characterized yet. To our
knowledge, up to now there is only a chamber study (Li et al., 2015) which has investigated
the evolution of aerosol particles emitted by wheat straw burning under dark conditions.
Although field studies (Adler et al., 2011; Liu et al., 2016) witnessed the evolution in mass
concentrations, size distribution, oxidation state and optical properties of aerosol particles
emitted by agricultural residues burning, these changes could be also influenced by other
emission sources and meteorological conditions as well. Since NMOGs emitted by agricultural
residues burning are not fully quantified, it is still challenging to predict the concentration and
physicochemical properties of SOA resulted from biomass burning (Spracklen et al., 2011;
Jathar et al., 2014; Shrivastava et al., 2015). Bruns et al. (2016) suggsted that the 22 major
NMOGs identified in residential wood combustion could explain the majority of observed SOA,
but it remains unclear whether identified NMOGs emitted by agricultural residues burning
could fully (or at least largely) explain the SOA formed. In addition, aerosol mass spectrometry
(AMS) has been widely used to characterize sources and evolution of ambient OA (Zhang et
al., 2011). Although agricultural residues burning is an important type of biomass burning in
Asia and especially in China, the lack of AMS spectra for primary and aged OA from
agricultural residues burning significantly limits further application of AMS in BBOA research.

In this study, plumes from agricultural residues open burning were directly introduced into

a large indoor chamber to firstly characterize primary emissions and then investigate their
photochemical evolution under ~25°C and ~50% relative humidity. Corn, rice and wheat straws,
which accounts for more than 90% of the crop residues burned in China (FAO, 2017), were



chosen. A suite of advanced online and offline techniques were utilized to measure gaseous and
particulate species, enabling comprehensive measurements of emission factors of gaseous and
particulate compounds for burning of each types of straw under ambient dilution conditions.
In addition, corresponding formation and transformation of SOA during photochemical aging
was investigated using a large indoor smog chamber. This work would help improve our
understanding of primary emission, SOA formation and thus environmental impacts of
agricultural residues burning.
**2 Materials and methods**
**2.1 Experimental setup**
Photochemical aging was investigated in a smog chamber in the Guangzhou Institute of
Geochemistry, Chinese Academy of Sciences (GIG-CAS). The GIG-CAS smog chamber is a
~30 m$^3$ fluorinated ethylene propylene (FEP) reactor housed in a temperature-controlled room.
Details of the chamber setup and associated facilities are provided elsewhere (Wang et al., 2014;
Liu et al., 2015, 2016; Deng et al., 2017). Briefly, 135 black lamps (1.2 m long, 60 W Philips,
Royal Dutch Philips Electronics Ltd, the Netherlands) are used as light sources, giving a $NO_2$
photolysis rate of approximately 0.25 min$^{-1}$. Two Teflon-coated fans are installed inside the
reactor to ensure introduced gaseous and particulate species mixed well within 2 min. Prior to
each experiment, the reactor was flushed with purified dry air at a rate of 100 L min$^{-1}$ for at
least 48 h.
Corn, rice and wheat straws were collected from Henan, Hunan and Guangdong province,
respectively. Since moisture content in straws would affect emission factors of atmospheric
pollutants (Sanchis et al., 2014; Ni et al., 2015), all the agricultural residues used in this study
were dried in a stove at 80 °C for 24 h before being burned. After baking, water content in the
crop residues was less than 1%. In each experiment, ~300 g straws were burned and the burning
typically lasted for 3-5 min. Straws were ignited by a butane-fueled lighter and burned under



open field burning conditions. The resulting smoke was collected by an inverted funnel and
introduced into the chamber using an oil-free pump (Gast Manufacturing, Inc, USA) at a flow
rate of ~15 L min$^{-1}$ through a 5.5 m long copper tube (inner diameter: 3/8 inch), and the
residence time in the tube was estimated to be <2 s. Before each experiment, the transfer tube
was pre-flushed for 15 min with ambient air and 2 min with smokes (not introduced into the
chamber reactor). During the whole process, the tube was heated at 80 °C to reduce the losses
of organic vapors. Based on the volumes of the smoke introduced and the chamber reactor, the
dilution ratios were estimated to be 1300-4000, falling into the typical range (1000-10000)
under ambient dilution conditions (Robinson et al., 2007). After being characterized in dark
for >20 min, black lumps were turned on and the diluted smoke were photochemically aged
for 5 h. At the end, black lamps were switched off and the aged aerosols were characterized in
the next one hour to correct the particle wall loss.

In total 20 experiments were conducted (9 for rice straw, 6 for corn straw and 5 for wheat

straw), among which 14 experiments were conducted only in the dark to measure primary
emissions and 6 experiments were carried out both in the dark and under irradiation to
investigate photochemical evolution of open straw burning emissions. Tables 1 and 2
summarize important experimental conditions and key results for all the experiments.
**2.2 Instrumentation**
Gaseous and particulate species were monitored with a suite of online and offline instruments.
Commercial instruments were used for online monitoring of $NO_x$ (EC9841T, Ecotech,
Australia), $NH_3$ (Model 911-0016, Los Gatos Research, USA) and $SO_2$ (Model 43i, Thermo
Scientific, USA). Volatile organic compounds (VOCs) were continuously measured using a
proton-transfer-reaction time-of-flight mass spectrometer (PTR-TOF-MS; Model 2000,
Ionicon Analytik GmbH, Austria). Calibration of the PTR-TOF-MS was performed every few
weeks using a certified custom-made standard mixture of VOCs (Ionicon Analytik Gmbh,





Austria) that were dynamically diluted to 6 levels (2, 5, 10, 20, 50 and 100 ppbv). Methanol,
acetonitrile, acetaldehyde, acrolein, acetone, isoprene, crotonaldehyde, 2-butanone, benzene,
toluene, o-xylene, chlorobenzene and α-pinene were included in the calibration mixture. Their
sensitivities, indicated by the ratio of the normalized counts per second to the concentration
levels of the VOCs in ppbv, were used to convert the raw PTR-TOF-MS signal to concentration
(Huang et al., 2016). Quantification of the compounds that were not included in the mixture
was performed by using calculated mass-dependent sensitivities based on the measured
sensitivities (Stockwell et al., 2015). Mass-dependent sensitivities were linearly fitted for
oxygen-containing compounds and the remaining compounds separately. The decay of toluene
measured by PTR-TOF-MS was used to derive the OH radical concentrations for every 2 min
during each experiment, and the OH exposure was calculated as the product of the OH
concentration and the time interval. Air samples were also collected from the chamber reactor
using 2-Liter electro-polished stainless-steel canisters before and after smoke injection. In total
67 $C_2$-$C_{12}$ NMHCs were measured (Table S1) using an Agilent 5973N gas chromatography
mass-selective detector/flame ionization detector (GC-MSD/FID; Agilent Technologies, USA)
coupled to a Preconcentrator (Model 7100, Entech Instruments Inc., USA), and analytical
procedures have been detailed elsewhere (Wang and Wu, 2008; Zhang et al., 2010; Zhang et
al., 2012). $CH_4$ and CO were analyzed using a gas chromatography (Agilent 6980GC, USA)
coupled with a flame ionization detector and a packed column (5A molecular sieve 60/80 mesh,
3 m × 1/8 in) (Zhang et al., 2012), and $CO_2$ was analyzed using a HP 4890D gas chromatograph
(Yi et al., 2007). The detection limits were all less than 30 ppbv for $CH_4$, CO and $CO_2$. The
relative standard deviations (RSDs) of CO and $CO_2$ measurements were both less than 3%
based on seven duplicate injection of 1.0 ppmv standards (Spectra Gases Inc, USA).
Particle number/volume concentrations and size distribution were measured with a
scanning mobility particle sizer (SMPS; Classifier model 3080, CPC model 3775, TSI



Incorporated, USA). The SMPS was operated with a sheath flow of 3.0 L min$^{-1}$ and a sampling
flow of 0.3 L min$^{-1}$, allowing for a size scanning range of 14 to 760 nm within 255 s. A high-
resolution time-of-flight aerosol mass spectrometer (HR-TOF-AMS; Aerodyne Research
Incorporated, USA) was used to measure chemical compositions of non-refractory aerosol
particles (DeCarlo et al., 2006). The instrument alternated every one min between the high
sensitivity V mode and the high resolution W mode. The toolkit Squirrel 1.57I was used to
obtain real-time concentration variations of sulfate, nitrate, ammonium, chloride and organics,
and the toolkit Pika 1.16I was used to determine the detailed compositions of OA (Aiken et al.,
2007; Aiken et al., 2008; Canagaratna et al., 2015). The AMS signal at m/z 44 was corrected
for the contribution from gaseous $CO_2$. Ionization efficiency of the AMS was calibrated
routinely by measuring 300 nm monodisperse ammonium nitrate particles. Considering the
underestimation of particulate matter by AMS, aerosol mass measured by AMS was corrected
with the data from the SMPS and the aethalometer. Conductive silicon tubes were used for
aerosol sampling to reduce electrostatic losses of particles.

BC was measured with a seven-channel aethalometer (Model AE-31, Magee Scientific,

USA). Cheng et al. (2016) measured the mass absorption efficiency (MAE) of BC from
biomass burning at wavelengths of 532 and 1047 nm respectively, and the absorption Ängström
exponents (AAE) were estimated to be in the range of 0.9-1.1. Based on relationship between
MAE and wavelength, a MAE value of 4.7 m$^2$ g$^{-1}$ was calculated for 880 nm by assuming the
AAE to be 1.0. The MAE value was then applied to convert absorption data in 880 nm to BC
mass concentrations. Aethalometer attenuation measurements were corrected for particle
loading effects and the scattering of filter fibers using the method developed by Kirchstetter
and Novakov (2007) and Schmid et al. (2006).





**2.3 Data analysis**
**2.3.1 Particle effective density**
Assuming that particles are spherical and non-porous, the effective density ($\rho_{eff}$) can be
estimated by Eq. (1) (DeCarlo et al. 2004; Schmid et al. 2007):
$$\rho_{eff} = \rho_0 \cdot \frac{d_{va}}{d_m} \qquad (1)$$

where $\rho_0$ is the standard density (1.0 g cm$^{-3}$), and $d_{va}$ and $d_m$ are the AMS-measured vacuum
aerodynamic diameter and SMPS-measured mobility diameter. The input diameters to this
equation were determined by comparing distributions of vacuum aerodynamic and electric
mobility diameters, using the AMS and SMPS respectively. Derived $\rho_{eff}$ was used to convert
volume concentrations of aerosol particles measured by the SMPS to mass concentrations.
**2.3.2 Emission factors and modified combustion efficiency**
The carbon mass balance approach (Ward et al., 1992; Andreae and Merlet, 2001) was used to
calculate fuel based emission factors (EF) for each compound (g kg$^{-1}$ dry fuel). The emission
factor for the $i$th species, EF$_i$, is calculated by Eq. (2):
$$EF_i = \frac{m_i \cdot EF_C}{\Delta[CO_2] + \Delta[CO] + \Delta[PM_C] + \Delta[HC]} \qquad (2)$$

where $m_i$ is the concentration (g m$^{-3}$) of the $i$th species; $\Delta[CO_2]$, $\Delta[CO]$, and $\Delta[HC]$ are the
background-corrected carbon mass concentration (g-C m$^{-3}$) of the $CO_2$, CO, and hydrocarbons,
respectively; $\Delta[PM_C]$ is the background-corrected carbon in the particle phase (g-C m$^{-3}$); $EF_C$
is the emission factor of carbon into the air determined by elemental analysis, given by Eq. (3):
$$EF_C = \frac{m_{fuel} \cdot \omega_{fuel} - m_{ash} \cdot \omega_{ash}}{m_{fuel}} \qquad (3)$$

where $\omega_{fuel}$ and $\omega_{ash}$ are mass fractions of carbon in the dry fuel and its ash, and $m_{fuel}$ and $m_{ash}$
are the mass of dry fuel and its ash. The modified combustion efficiency (MCE) is defined by
Eq. (4) (Heringa et al., 2011; Hennigan et al., 2011; Ni et al., 2015):
$$MCE = \frac{\Delta[CO_2]}{\Delta[CO_2] + \Delta[CO]} \qquad (4)$$





### 2.3.3 Ozone formation potential

The ozone formation potential (OFP) of NMHCs was calculated from the emission factor and

maximum incremental reactivity (MIR) of each individual NMHCs, using Eq. (5):

$$OFP = \sum_{i=1}^{n}(EF_i \cdot MIR_i) \qquad (5)$$

where OFP is the ozone formation potential of NMHCs emitted from per unit of biomass (unit:

g kg$^{-1}$), and MIR$_i$ is the MIR of the $i$th NMHC (unit: g O$_3$ per g NMHC) (Carter 2008).

### 2.3.4 Wall loss corrections

Due to the loss of particles and vapors to chamber walls, measured data in chamber studies

need to be corrected for wall loss. For this purpose, in our study one-hour dark decay of aged

aerosols was undertaken after photochemical aging was terminated. The loss of particles on the

chamber wall is a first-order process (McMurry and Grosjean, 1985). The wall-loss rates of

AMS-measured organics, sulfate, nitrate, chloride and ammonium were determined using the

dark decay data and were applied to wall-loss correction for the entire experiment. By assuming

that the condensed materials on the wall remains completely in equilibrium with the gas phase,

we used the $\omega=1$ case to correct the OA mass, where $\omega$ is a proportionality factor of organic

vapor partitioning to chamber walls and suspended particles (Weitkamp et al., 2007; Henry et

al., 2012). For SMPS measurements, the number concentration in each size channel (110

channels in total) was corrected for wall loss separately, since wall loss rates of aerosol particles

are size-dependent (Takekawa et al., 2003).

### 2.3.5 OA production prediction

In this study, twenty NMOGs which have been used to estimate SOA yields by previous work

(Ng et al., 2007b; Chan et al., 2009; Hildebrandt et al., 2009; Gómez Alvarez et al., 2009; Chan

et al., 2010; Shakya and Griffin, 2010; Chhabra et al., 2011; Nakao et al., 2011; Borras and

Tortajada-Genaro, 2012; Yee et al., 2013; Lim et al., 2013) were identified using PTR-TOF-

MS, and the applied SOA yields are summarized in Table S2. The mass concentration of SOA



([SOA]$_{predicted}$, µg m$^{-3}$) formed from these twenty precursors can be estimated using Eq. (6):
$$[SOA]_{predicted} = \sum_i (\Delta[X_i] \cdot Y_i) \qquad (6)$$
where $\Delta[X_i]$ (µg m$^{-3}$) is the reacted amount of the $i$th gas-phase precursor and $Y_i$ is the
corresponding SOA yield.
Assuming that primary OA (POA) levels kept constant during aging processes, the mass
concentration of SOA formed could be estimated as the difference in OA mass concentrations
before and after photochemical aging. It should be noted that POA would decrease during aging
processes (Tiitta et al., 2016), probably leading to the underestimation of the formed SOA. In
papers where those SOA yields were borrowed from, no organic vapor wall loss were
accounted for when calculating the mass concentration of the formed SOA, so the same wall
loss correction method was used when comparing the predicted SOA and the formed SOA.
**3 Results and discussion**
**3.1 Emissions of gaseous pollutants**
Table 1 compares emission factors of gaseous and particulate species measured in our and
previous studies. In our study, emission factors of NO$_x$ were 1.47±0.61, 5.00±3.94, and
3.08±0.93 g kg$^{-1}$ for rice, corn and wheat straw, and NO accounted for 84±11% of NO$_x$ primary
emission for all experiments. Emission factors of NH$_3$ were measured to be 0.45±0.15,
0.63±0.30 and 0.22±0.19 g kg$^{-1}$ for rice, corn and wheat straw. Our measured emission factors
of reactive nitrogen species were comparable to those reported by previous studies (Li et al.
2007; Tian et al. 2011). Emission factors of SO$_2$ were 0.07±0.07, 0.99±1.53 and 0.72±0.34 g
kg$^{-1}$ for rice, corn and wheat straw. Our measured emission factors of SO$_2$ were lower than
those reported by Cao et al., (2008) and Kim Oanh et al. (2015) for rice straw, but higher those
reported by Cao et al., (2008) for corn and wheat straw. Due to low sulfur contents in crop
straws, the SO$_2$ emission factors for open burning of crop residues were much lower than those
for coal combustion, which were determined to be 9.92±2.83 g kg$^{-1}$ for household combustion



of coal cakes (Ge et al., 2004).
Emission factors of NMHCs were 5.04±2.04, 2.47±2.11 and 3.08±2.43 g kg$^{-1}$ for rice,
corn and wheat straw, respectively (Table 1). Our results were higher than those reported by
previous studies (Li et al., 2009; Wang et al., 2014), partly due to the fact that more NMHCs
were analyzed in our study (67 species in total). As shown in Figure 1a-c, olefins and acetylene
accounted for 56-58% of the total NMHCs, followed by alkanes (22-28%) and aromatic
hydrocarbons (16-21%). Table S1 and Figure 2 show the emission factors of each NMHC for
open burning of different straws. Emission factors of unsaturated hydrocarbons ranged from
1.37 (corn) to 2.91 g kg$^{-1}$ (rice), with the majority being ethene, acetylene and propene.
Emission factors of alkanes ranged from 0.69 (corn) to 1.09 g kg$^{-1}$ (rice), with ethane and
propane being the two most abundant compounds. The emission factors of aromatic
hydrocarbons were in the range of 0.42 (corn) to 1.04 (rice), and benzene and toluene are
dominant species. It is worth noting that major compounds in the three groups (alkanes, alkenes
and aromatic hydrocarbons) were all negatively correlated with the modified combustion
efficiency (Figure S1), suggesting that more efficient combustion would reduce their emissions.
Based on their emission factors, we calculated the ozone formation potential for each
NMHC. The summed ozone formation potential were 22.5±10.1, 13.7±12.4 and 16.3±13.5 g
kg$^{-1}$ for open burning of rice, corn and wheat straw, respectively. As shown in Figure 1d-e, the
relative contributions of olefins to the total ozone formation potential could reach >80%.
Ethene was the largest ozone precursor (35-42%), followed by propene (16-28%), and these
two compounds contributed 58-64% of the total ozone formation potential. Although the
emission factors of aromatic hydrocarbons were lower than those of alkanes, their ozone
formation potential was dominant over those of alkanes, with toluene being the largest
contributor among all the aromatic hydrocarbons. The contribution of alkanes to the total ozone
formation potential was minor (2-3%).





### 3.2 Emission of particulate matters


The emission factors of particulate matters were 3.73±3.28, 5.44±3.43, 6.36±2.98 g kg$^{-1}$ for
rice, corn and wheat straw, lower than those reported in the previous studies (Table 1). As
suggested by Robinson et al. (2007), the POA emission factors would decrease with increasing
dilution ratios, due to evaporation of semi-volatile organic compounds. In this study, the
dilution ratios ranged from 1300 to 4000, which were within the typical range of ambient
dilution ratios (1000-10000) (Robinson et al. 2007). Therefore, it can be expected that emission
factors of primary organic carbon (POC) measured in our study (2.05-4.11 gC kg$^{-1}$) were lower
than those measured by previous work with dilution ratios of 5-20 (Li et al. 2007; Ni et al.
2015). Moreover, it has been shown that the modified combustion efficiency could affect
emission factors (Heringa et al., 2011; Stockwell et al., 2015). Figure S2 shows negative
correlations of the modified combustion efficiency with emission factors of PM and POC
($p$<0.05 for both cases), indicating that enhancement of combustion efficiency could reduce
the emissions of PM and POC. In our study, all straws were pre-baked to reduce the moisture
content to <1%, and this treatment could increase the modified combustion efficiency and thus
reduce emission factors of particulate matters (Ni et al., 2015). In addition, the amount of straws
burned each time in our experiments was much less than that in the fields, which is expected
to avoid oxygen deficit during burning to some extent and thus increase the modified
combustion efficiency as well.
While POA emission factors showed large variability for different types of straw, BC
emission factors were relatively constant (0.22-0.27 gC kg$^{-1}$). Since BC is a mixture of non-
volatile compounds in particulate matters, as expected, its emission factors measured in our
work were comparable to those reported under lower dilution conditions (Li et al. 2007; Ni et
al. 2015). The Δ[POA]/Δ[CO] ratios ranged from 0.022 to 0.133 in our study, larger than those
(0.001-0.067) measured in chamber studies for hard- and soft-wood fires (Grieshop et al.,



2009b) and vegetation commonly burned in North American wildfires (Heringa et al., 2011),
but lower than those (0.051-0.329) obtained in field campaigns (Jolleys et al., 2012).
For particle numbers, the emission factors were $(2.94\pm0.91)\times10^{15}$, $(7.29\pm4.17)\times10^{15}$,
$(5.87\pm2.89)\times10^{15}$ particle kg$^{-1}$ for rice, corn and wheat straw, respectively (Table 1). Our results
were comparable to that $(1\times10^{15}$ particle kg$^{-1})$ for crop residues burning (Andreae and Merlet,
2001) and those $(3.2\times10^{15}$-$10.9\times10^{15}$ particle kg$^{-1})$ for wood burning (Hosseini et al., 2013) but
two magnitudes larger than those for crop residues burning in a sealed stove (Zhang et al. 2008).
**3.3 Evolution of particles**
**3.3.1 Growth of particle size**
Figure 3 shows the evolution of particle size distribution after photochemical aging of 0, 0.5,
2.5 and 5 h. Aerosol particles emitted from open straw burning were peaked at 40-80 nm under
ambient dilution conditions. The geometric mean diameters for primarily emitted particles in
this study were smaller than those (100-150 nm) reported for crop residuals burning under low
dilution conditions (Zhang et al., 2011; Li et al., 2015), probabley due to evaporation of organic
vapors under the high dilution conditions (Lipsky et al., 2006) and coagulation of fine particles
under the low dilution conditions (Hossain et al., 2012).
After switching on black lamps, apparent growth of particle size was observed. In all the
aging experiments, growth rates of particle diameters in the first 0.5 h were 10 times larger
than those afterwards, and after 5 h aging the geometric mean diameters peaked at 60-120 nm.
For instance, in the photochemical aging experiment for wheat straw burning (Figure 3c), the
growth rate of particles was 18 nm h$^{-1}$ in the first 0.5 h and decreased to ~1 nm h$^{-1}$ during the
following 4.5 h. The size distribution of aged aerosol particles in our study is similar to those
of ambient particles under the severe biomass burning impact during haze events (Betha et al.,
2014; Niu et al., 2016).





### 3.3.2 Particle mass enhancement

Figure 4 shows the chemical evolution of aerosol particles during the 5 h photochemical aging of wheat straw burning. During the whole process, OA kept increasing and was dominant over inorganic species. After 3 h of photochemical aging, the levels of all the inorganic species were constant, and nitrate was the second most abundant component with a mass fraction of 7%, followed by chloride (2%), ammonium (1%) and sulfate (<1%). Figure 4b depicts [OA] evolution as a function of OH exposure. OA increased slowly at the first ~0.2 h, and then increased rapidly with OH exposure.

The OA enhancement ratio, defined as the mass ratio of aged OA at the end of each aging experiment to POA, was calculated. In the six aging experiments, the OH exposure and OA enhancement ratios ranged from $(1.87\text{-}4.97)\times10^{10}$ molecule cm$^{-3}$ s and 2.4-76, respectively. Assuming an average OH concentration of $1.5\times10^6$ molecule cm$^{-3}$ in the ambient air (Hayes et al., 2013), this means that rapid SOA formation would occur in 3.5-9.2 h during the daytime after straw burning. The OA enhancement ratios determined in our study were higher than those (0.7-2.9) for the combustion of vegetation commonly burned in North American wildfires (Hennigan et al., 2011), and comparable to those (0.7-6.9) for wood burning (Grieshop et al., 2009b; Heringa et al., 2011).

Recently, Bruns et al., (2016) found that 22 NMOGs emitted from residential wood burning could explain the majority of the formed SOA. In our study, 20 of the 22 NMOGs were detected and quantified with the PTR-TOF-MS. Concentration differences of each compound before and after photo-oxidation were calculated to estimate the SOA formed from these precursors. Since SOA formation highly depends on oxidation conditions, SOA yields for a certain precursor vary with VOC/NO$_x$ ratios. In our work, we chose a set of SOA yields for these NMOGs based on the observed VOC/NO$_x$ ratio in the chamber experiments. More specifically, if the observed VOC/NO$_x$ ratio for a certain precursor in the chamber was within





the VOC/$NO_x$ range reported in literature, the mean value of the highest and lowest yields
within the VOC/$NO_x$ range in literature was used to estimate the SOA formed from the
precursor in the chamber; if the observed VOC/$NO_x$ ratio for a certain precursor was higher
than the maximum VOC/$NO_x$ ratio reported in literature, we chose the yield reported at the
maximum VOC/$NO_x$ ratio; if the observed VOC/$NO_x$ ratio was lower than the minimum
VOC/$NO_x$ ratio reported in literature, we chose the yield reported at the minimum VOC/$NO_x$
ratio.

Figure 5a shows the time series of POA, $SOA_{predicted}$ and unexplained SOA in a typical

aging experiment. The contribution of $SOA_{predicted}$ by the 20 NMOGs was minor, and large
fractions of observed SOA could not be explained. In all the experiments, only 5.0-27.3% of
the observed SOA mass could be explained by the 20 NMOGs (Figure 5b). Even if the highest
SOA yield for each precursor reported in literature was used, 60-90% of observed SOA mass
still could not be explained. It has been suggested that aqueous-phase oxidation of alkenes
could produce substantial SOA (Ervens et al., 2011). Considering large emissions of olefins
from straw burning (Figure 1a-c), we also estimated the SOA formed from the three most
abundant alkenes (ethene, acetylene, and propene) with their newly-developed SOA yields (Ge
et al., 2016; Jia and Xu, 2016; Ge et al., 2017), and their total contribution to the observed SOA
was found to be negligible (<0.5%). Therefore, there are still unknown precursors and/or
physicochemical processes contributing the majority of SOA formed from open straw burning.
**3.3.3 OA mass spectrum evolution**
In the high resolution W mode of AMS, ions generated from particles could be identified by
their exact mass-charge ratio (m/z) and then grouped into CHON, CHO, CHN and CH families.
Figure 6 presents the evolution of OA mass spectra. For POA (Figure 6a), CH-family was the
major component with a mass fraction of 68%, followed by CHO (23%), CHN (6%), and
CHON (2%). The ions at m/z 43, 41 and 55 were the dominant peaks in the POA mass spectrum.



The major ions at m/z 27, 39, 41, 55, 57, 67 and 69 belonged to the CH-family and could be
the fragments of hydrocarbons (Weimer et al., 2008). The peaks at m/z 28, 29, 43, 44 and 55
contained considerable CHO ions, and the corresponding ions ($CO^+$, $CHO^+$, $C_2H_3O^+$, $CO_2^+$ and
$C_3H_3O^+$) could be the fragments of aldehydes, ketones and carboxylic acid (Ng et al., 2011a).
The peak at m/z 91 was mainly attributed to $C_7H_7^+$, possibly originating from aromatic
compounds.

The mass spectra of aged OA was quite different from that of POA (Figure 6b-c). The mass

fraction of the CH-family decreased to 46% and was comparable to that of CHO-family, while
the contribution of N-containing OA (CHN and CHON) increased to ~11%. The ions at m/z 44
and 43, mainly coming from the CHO-family, became the dominant peaks for the aged OA.
The fractions of two major masses at m/z 44 ($f_{44}$) and m/z 43 ($f_{43}$) in OA can be used to generate
an $f_{44}$ vs. $f_{43}$ triangular space, in which oxygenated organic aerosol (OOA) moves towards the
apex during the aging process (Ng et al., 2010). In addition, $f_{44}$ in the ambient air was suggested
to be 0.07±0.04 for semi-volatile OOA (SV-OOA) and 0.17±0.04 for low-volatility OOA (LV-
OOA), respectively (Ng et al., 2010). Figure 7a plots $f_{44}$ and $f_{43}$ of the POA and the aged OA
in all the six experiments. Most of data are within the $f_{44}$ vs. $f_{43}$ triangular space and close to
the left margin. Photochemical aging led to increase in $f_{44}$ for all the experiments, suggesting
transformation of OA from SV-OOA to LV-OOA. For comparison, the $f_{43}$ did not change
significantly in all the experiments. The main ions at m/z 43 were $C_2H_3O^+$ and $C_3H_7^+$. It can be
observed in Figure 6c that the increased contribution of $C_2H_3O^+$ and the decrease contribution
of $C_3H_7^+$ were comparable during photoreaction.

The ion at m/z 60, mainly consisting of $C_2H_4O_2^+$, is regarded as a BBOA marker, and the

mass fraction of this ion in OA, $f_{60}$, is widely used to probe the evolution of BBOA (Brito et
al., 2014; May et al., 2015). Figure 7b plots evolution of $f_{44}$ and $f_{60}$ in all the experiments
conducted in this study, in order to compare with measurements in aging biomass burning



plumes (Cubison et al., 2011) and those in the POA from different types of biomass burning
(Alfarra et al., 2007; Brito et al., 2014; May et al., 2015). Photo-oxidation caused increase in
$f_{44}$ and decrease in $f_{60}$, and this is consistent with the general evolution of OA in ambient
biomass burning plumes (Cubison et al., 2011). However, our measured $f_{60}$, 0.003-0.006 in the
POA from open straw burning and 0.002-0004 in aged OA, were all lower than those from
other field campaigns and quite near the background $f_{60}$ level of 0.003 for ambient OA (Cubison
et al., 2011; Figure 7b). Low values of $f_{60}$ (0.005-0.02) were also reported by Hennigan et al.
(2011) in a chamber study for fuels commonly burned in wildfires. In their study, biomass
burning took place in a 3000 $m^3$ combustion chamber, and the smokes were then injected into
another chamber for aging experiments with a dilution ratio of ~25. Previous studies have
demonstrated that levoglucosan is a semi-volatile compound with a saturation concentration of
~8 μg $m^{-3}$ at 293 K (Grieshop et al., 2009b; Huffman et al., 2009; Hennigan et al. 2011). As a
result, high dilution conditions used in our study would cause levoglucosan to evaporate, and
this may at least partly explain the low $f_{60}$ observed in the POA from straw burning. From
previous studies, the levoglucosan/OC ratios of straw burning ranging from 4.92 to 16.8% (4
types of vegetation summarized; Dhammapala et al., 2007; Kim Oanh et al., 2011; Hall et al.,
2012) were not significantly (two-sample t-test, $p>0.05$) lower than those of prescribed fuel
burning, wildfire and wood burning ranging from 1.46 to 13.5% (20 types of vegetation
summarized; Hosseini et al., 2013; Shahid et al., 2015). So the difference in fuel type cannot
explain the lower $f_{60}$ observed in our study.
**3.3.4 Elemental ratio and oxidation state of OA**
In this study, the O/C and H/C ratios in the POA from different straws burning were in the
range of 0.20-0.38 and 1.58-1.74, respectively. After 5 h aging, O/C increased and H/C
decreased (Table 2). Kroll et al. (2011) proposed a metric, the average carbon oxidation state
($OS_c$), to describe the degree of oxidation of atmospheric organic species. $OS_c$ could be



calculated from the elemental composition of OA measured by AMS, given by Eq. (7):
$$OS_c = 2 \times O/C - O/H \qquad (7)$$
In this study, the $OS_c$ values for the fresh POA from open straw burning ranged from -
1.25 to -0.89, consistent with those suggested for BBOA (-1 to -0.7) (Kroll et al. 2011). During
photochemical aging, the $OS_c$ values increased linearly ($p<0.001$) with OH exposure (Figure
8), and the slopes were quite near each other even for different types of straws, implying AMS
measured $OS_c$ might be a good indicator of OH exposure and thereby of photochemical aging.
Figure 9 shows the Van Krevelen diagram of OA. In this study, the slopes of linear
correlations between H/C and O/C range from -0.49 to -0.24 for the five experiments. Slopes
of -1, 0.5 and 0 in the Van Krevelen diagrams indicate addition of carboxylic acids without
fragmentation, addition of carboxylic acids with fragmentation, and addition of
alcohols/peroxides, respectively (Heald et al., 2010; Ng et al., 2011a). Therefore, the slopes
determined in our study suggest that open straw burning OA aging resulted in net changes in
chemical composition equivalent to addition of carboxylic acid groups with C-C bond breakage
and addition of alcohol/peroxide functional groups.
**4 Conclusion**
In this study, primary emissions of open burning of rice, corn and wheat straw and their
photochemical were investigated using a large indoor chamber. Emission factors of $NO_x$, $NH_3$,
$SO_2$, 67 NMHCs, PM and particle number were measured under dilution ratios ranging from
1300 to 4000. Emission factors of PM (3.73 to 6.36 g $kg^{-1}$) and POC (2.05 to 4.11 gC $kg^{-1}$)
were lower than those reported in previous studies conducted at lower dilution ratios, probably
due to the evaporation of semi-volatile organic compounds. Emission factors of POC, PM and
major NMHCs compounds were all negatively correlated with the modified combustion
efficiency, suggesting that incomplete burning of agricultural residues could lead to larger
primary emission. Photochemical aging of primary emissions was investigated with OH




exposure equal to 3.2-9.2 hours under typical ambient conditions, and at the end of experiments
the OA mass concentrations increased by a factor of 2.4-7.6, suggesting that SOA could be
rapidly produced within several hours. Our estimation suggests that phenols are the most
important identified SOA precursors, and more than 70% of the formed OA still cannot be
explained by the oxidation of known precursors. Measurements using HR-TOF-AMS reveal
that after photochemical aging, signals for oxygen- and nitrogen-containing compounds were
largely increased, with $OS_c$ increased in a highly significant linear way with OH exposure.

**Acknowledgements**
This study was supported by Strategic Priority Research Program of the Chinese Academy of
Sciences (Grant No. XDB05010200), National Natural Science Foundation of China (Grant
No. 41530641/41571130031/41673116), National Key Research and Development Program
(2016YFC0202204) and Guangzhou Science Technology and Innovation Commission

(201505231532347).

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





**Table 1.** Primary emission factors measured for agricultural residues burning. All the units are
g kg$^{-1}$, except the unit for particle number (PN) is $10^{15}$ particle kg$^{-1}$. MCE: modified combustion
efficiency; NMHCs: non-methane hydrocarbons; POA: primary organic aerosol; POC primary
organic carbon; BC: black carbon.

| Species | Rice This study (n=9) | Others | Corn This study (n=6) | Others | Wheat This study (n=5) | Others |
|---|---|---|---|---|---|---|
| MCE | 0.926±0.049 | | 0.953±0.019 | | 0.949±0.035 | |
| $CO_2$ | 1262±81 | | 1477±28 | | 1423±60 | |
| CO | 63.5±41.4 | | 46.1±19.2 | | 48.6±33.0 | |
| $NO_x$ | 1.47±0.61 | 3.51±0.38[a] | 5.00±3.94 | 4.3±1.8[b] | 3.08±0.93 | 3.3±1.7[b]; 2.27±0.04[a] |
| $NH_3$ | 0.45±0.15 | 0.95±0.65[a]; 4.10±1.24[c] | 0.63±0.30 | 0.68±0.52[b] | 0.22±0.19 | 0.37±0.14[b]; 0.21±0.14[a] |
| $SO_2$ | 0.07±0.07 | 0.18±0.31[d]; 0.37±0.27[e]; 1.27±0.35[a] | 0.99±1.53 | 0.04±0.04[d] | 0.72±0.34 | 0.04±0.04[d]; 0.73±0.15[a] |
| NMHCs | 5.04±2.04 | 1.25[f] | 2.47±2.11 | 1.59±0.43[g] | 3.08±2.43 | 1.69±0.58[g]; 0.90[f] |
| PM | 3.73±3.28 | 8.5±6.7[h]; 8.3±2.2[e]; 13.2±1.44[i]; 4.2[c] | 5.44±3.43 | 12.2±5.4[h]; 11.7±1.0[b]; 5.36±0.55[i] | 6.36±2.98 | 11.4±4.9[h]; 7.6±4.1[b]; 5.30±0.30[i] |
| PN | 2.94±0.91 | 0.018±0.001[j] | 7.29±4.17 | 0.017±0.001[j] | 5.87±2.89 | 0.010±0.001[j] |
| POA | 2.99±1.00 | | 3.99±2.68 | | 5.96±0.19 | |
| POC | 2.05±0.72 | 3.3±2.8[h]; 6.02±0.60[i] | 2.52±1.66 | 6.3±3.6[h]; 3.9±1.7[b]; 2.06±0.34[i] | 4.11±0.29 | 5.1±3.0[h]; 2.7±1.0[b]; 2.42±0.13[i] |
| BC | 0.22±0.11 | 0.21±0.13[h] | 0.24±0.09 | 0.28±0.09[h]; 0.35±0.10[b] | 0.27±0.07 | 0.24±0.12[h]; 0.49±0.12[b] |

[a] Stockwell et al., 2015; [b] Li et al., 2007, PM correspond to $PM_{2.5}$; [c] Christian et al., 2003; [d] Cao et al., 2008; [e] Kim
Oanh et al., 2015, PM correspond to $PM_{2.5}$; [f] Wang et al., 2014, 56 NMHCs species summarized; [g] Li et al., 2009,
52 NMHCs species summarized; [h] Ni et al., 2015, PM correspond to $PM_{2.5}$; [i] Li et al., 2017, PM correspond to
$PM_1$; [j] Zhang et al., 2008.





**Table 2.** Overview of important experimental conditions and key results in the photochemical
oxidation experiments. The unit for OH exposure is $10^{10}$ molecule cm$^{-3}$ s. NA: data was not
available because no data was recorded in the W-mode.

| NO. | Straw type | Temp (°C) | RH (%) | OH exposure | POA | | | Aged OA | | | OA ER |
|---|---|---|---|---|---|---|---|---|---|---|---|
| | | | | | O/C | H/C | OS$_c$ | O/C | H/C | OS$_c$ | |
| Burn 1 | Rice | 25.0±0.4 | 48.9±1.4 | 3.80 | NA | NA | NA | NA | NA | NA | 2.7 |
| Burn 2 | Rice | 25.1±0.4 | 55.0±2.3 | 4.97 | 0.25 | 1.74 | -1.25 | 0.50 | 1.65 | -0.65 | 7.6 |
| Burn 3 | Corn | 25.5±0.4 | 53.0±2.9 | 4.16 | 0.38 | 1.66 | -0.89 | 0.60 | 1.66 | -0.46 | 3.6 |
| Burn 4 | Corn | 26.1±0.4 | 48.4±2.2 | 4.16 | 0.30 | 1.58 | -0.97 | 0.65 | 1.57 | -0.26 | 4.6 |
| Burn 5 | Wheat | 25.3±0.5 | 52.8±2.2 | 3.20 | 0.20 | 1.66 | -1.25 | 0.50 | 1.56 | -0.55 | 2.4 |
| Burn 6 | Wheat | 25.2±0.4 | 55.1±2.7 | 1.87 | 0.26 | 1.71 | -1.20 | 0.53 | 1.66 | -0.61 | 6.6 |








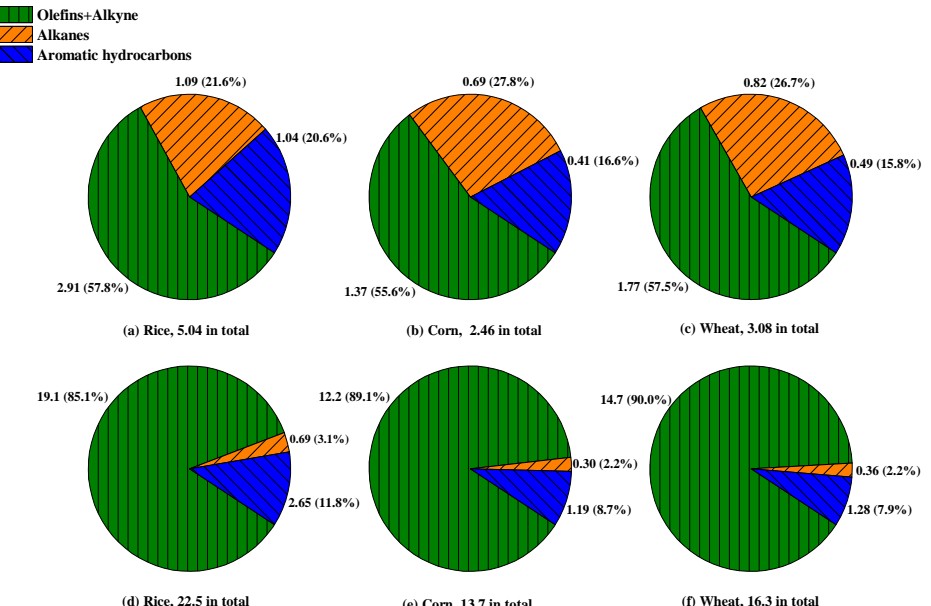

**Figure 1.** (a-c) Non-methane hydrocarbon (NMHC) compositions and (d-f) their relative
contribution to ozone formation potential (OFP) for open burning of rice, corn and wheat straw.




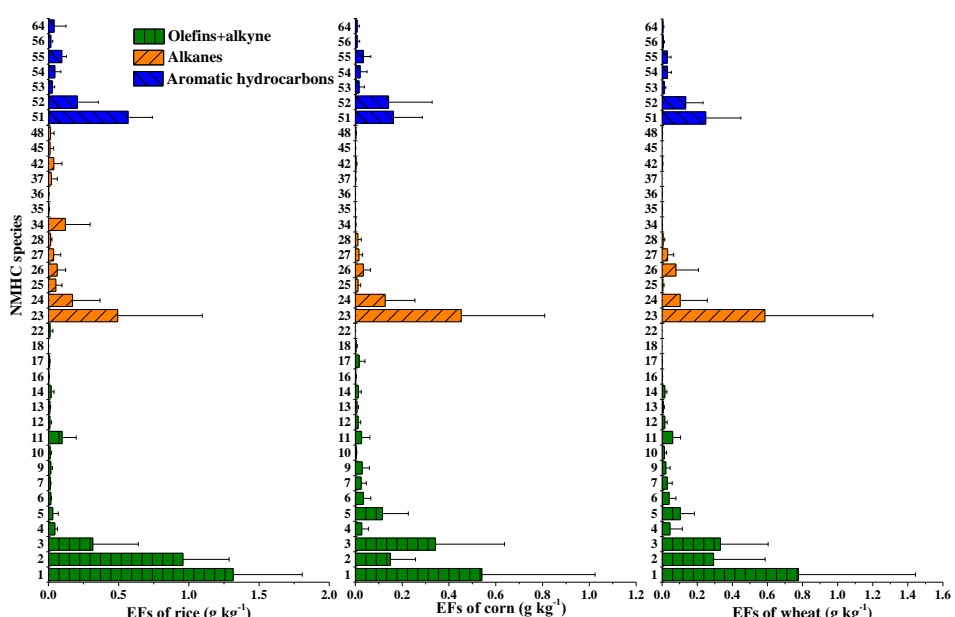


**Figure 2.** Emission factors (EFs) of NMHCs for straw burning of rice, corn and wheat. Only
species with emission factors >0.01 g kg$^{-1}$ are shown. The order of NMHC species is the same
as Table S1 in which a comprehensive dataset of emission factors measured in this work is
included.





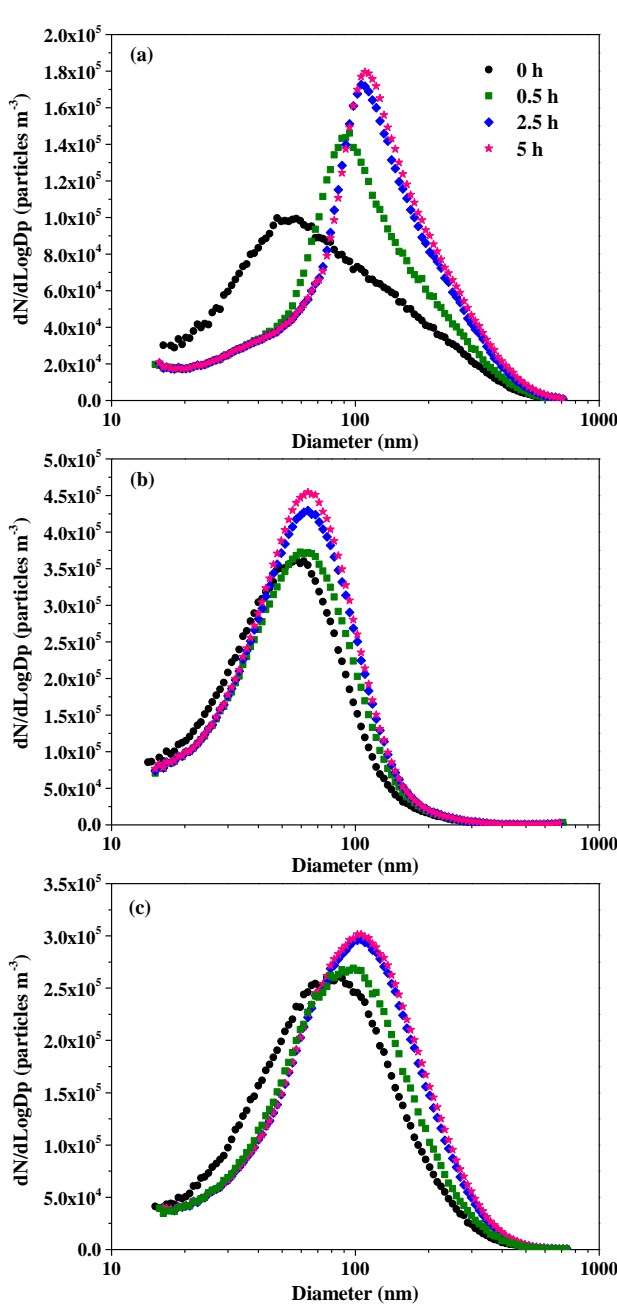


**Figure 3.** Particle size distributions in different burning. (a) Burn 2: rice straw; (b) Burn 3:

corn straw, (c) Burn 5: wheat straw.



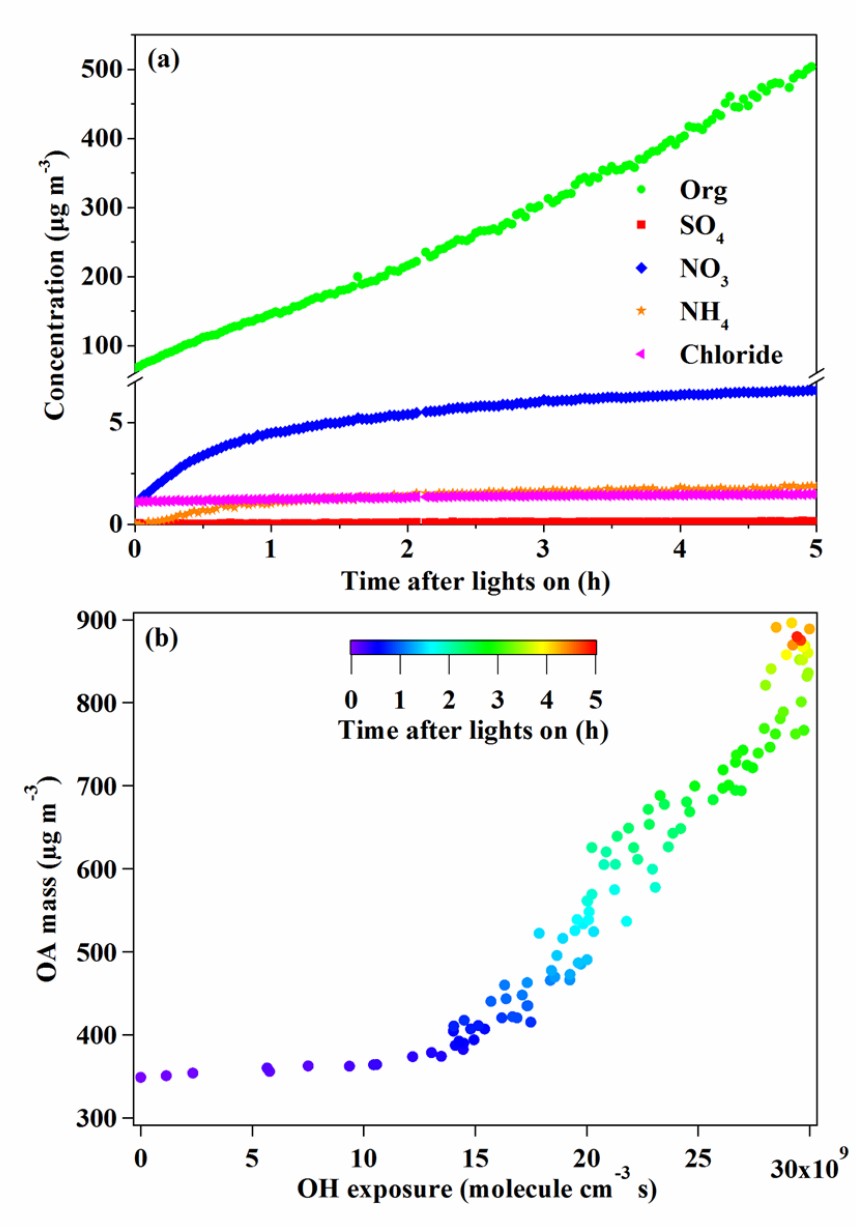


**Figure 4.** (a) The evolution of particulate matter components (Burn 2). (b) OA mass growth as
a function of OH exposure (Burn 5).

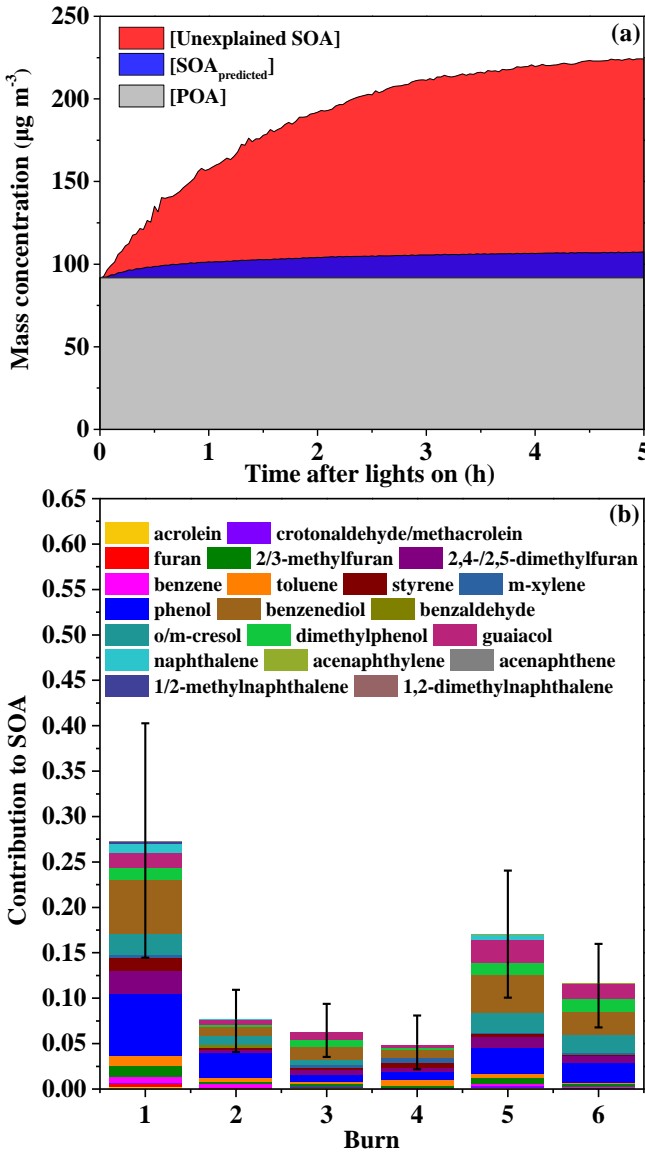

**Figure 5.** (a) Time series plots of concentrations of POA, secondary organic aerosol that can

be explained by the reacted precursors (SOA_predicted), the difference between the formed SOA

and the predicted SOA (Unexplained SOA) in Burn 6. (b) Contribution of 20 NMOGs to the

formed SOA at the end of photoreactions. Error bars correspond to the range of contributions

when the lowest/highest SOA yields in references were used for all precursors.





1027

**Figure 6.** (a) Mass spectrum of POA; (b) mass spectrum of aged OA; (c) Difference in mass

spectra between aged OA and POA. The data were all taken from Burn 5.



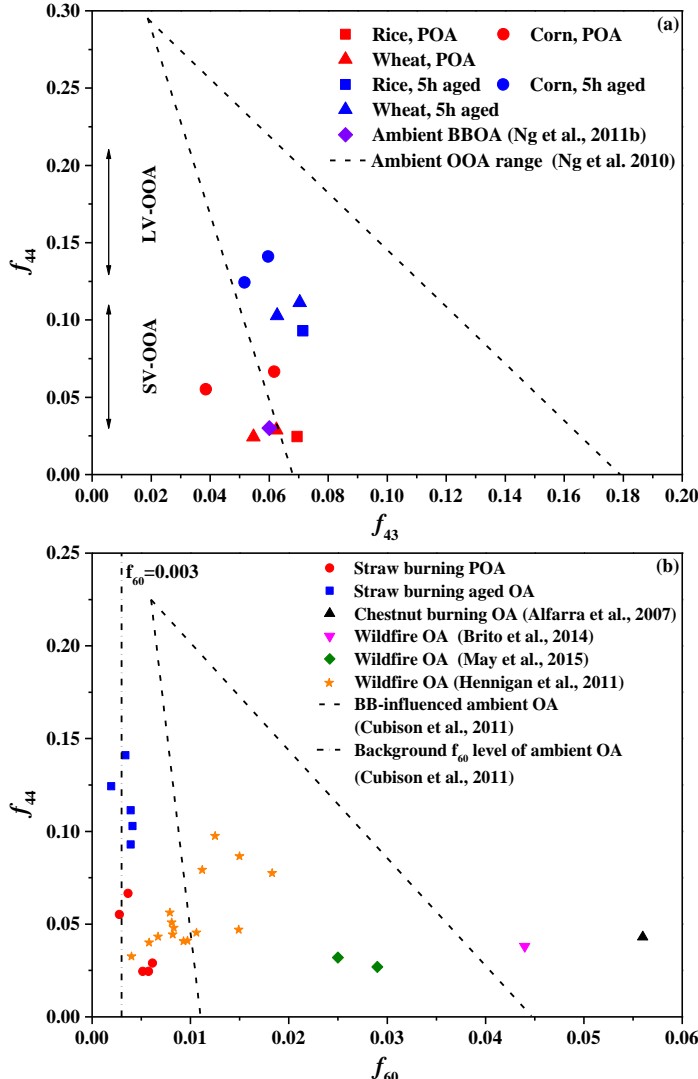

1030

**Figure 7.** (a) Comparison of $f_{44}$ vs $f_{43}$ determined in our work with those for the ambient BBOA

data sets (Ng et al., 2011b) and the ambient OOA range (Ng et al., 2010). The typical $f_{44}$ ranges

of ambient SV-OOA and LV-OOA are indicated with the vertical arrows. (b) Comparison of

$f_{44}$ vs $f_{60}$ for straw burning OA with those for other types of biomass burning OA (Alfarra et

al., 2007; Hennigan et al., 2011; Cubison et al., 2011; Brito et al., 2014; May et al., 2015).





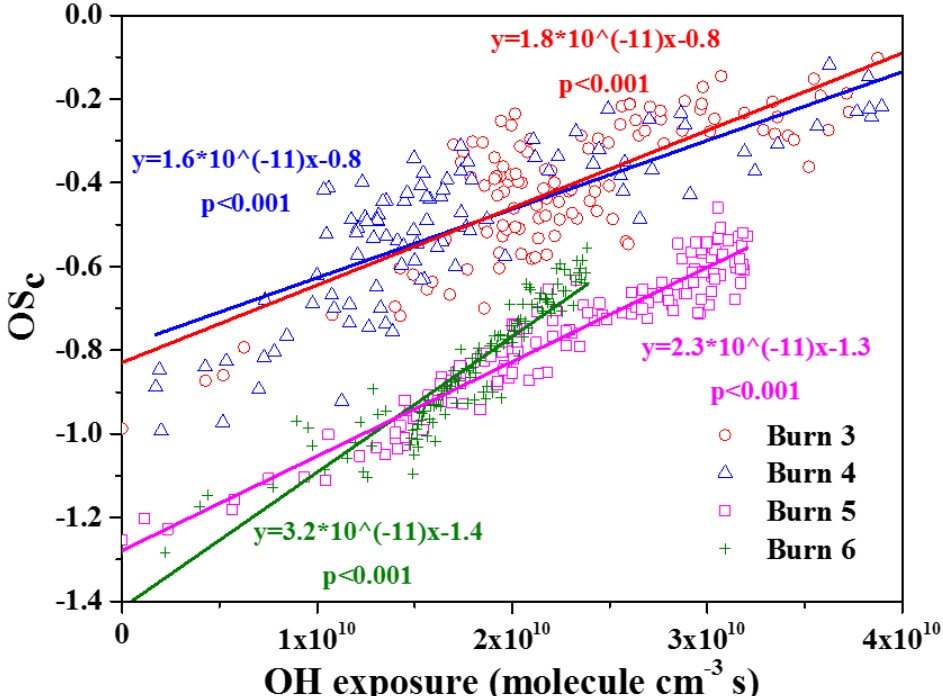


**Figure 8**. The growth of OA carbon oxidation state with OH exposure for burning corn (Burn
3 and 4) and wheat (Burn 5 and 6) straws. Data for burning rice straws were not included since
in Burn 1 AMS was then not run in W-mode.





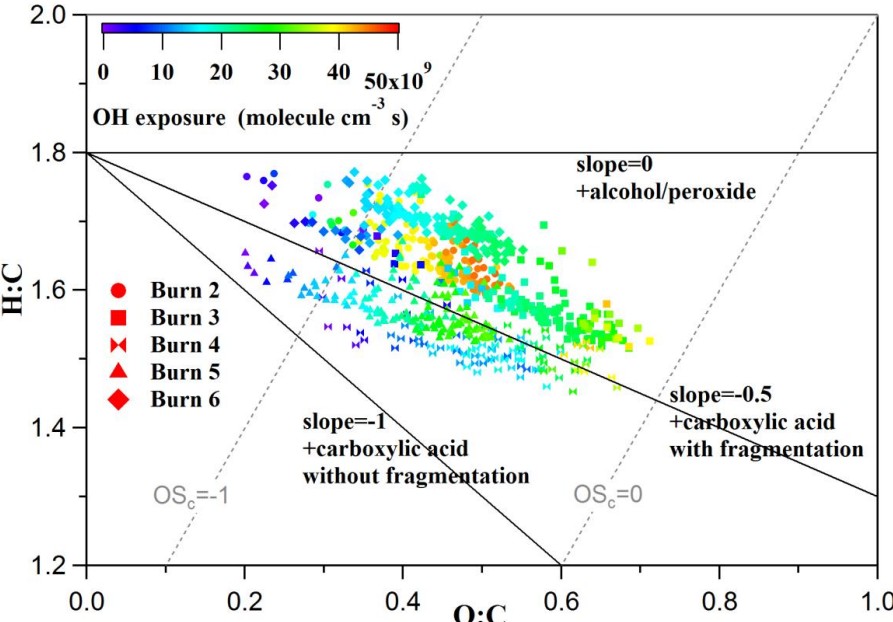


**Figure 9.** Van Krevelen diagram for the OA. Each slope corresponds to the addition of a
specific functional group to an aliphatic carbon.