# Peer review of "Open burning of rice, corn and wheat straws: primary"

_Atmospheric Chemistry and Physics, 2017_

## Referee Comment (RC1) · Anonymous Referee #1 · 8 Sep 2017

This manuscript investigated the emissions of various primary pollutants and photochemical evolution from burning three types of agricultural residues (corn, rice and wheat straws) by using a 30m3-smog chamber. The experimental design is reasonable, the results are reliable, and the conclusions are convincing. Considering the rare information on primary emissions and photochemical evolution of agricultural residues burning, the original data presented in this manuscript are very important for comprehensibly understanding the impact of the burning on the air quality, especially in China. The manuscript is well organized, and hence this reviewer recommends the manuscript be published in the journal. Specifics: Both biomass burning and domestic coal combustion have been recognized to make evident contribution to deteriorating

regional air quality especially in North China. If the authors had compared with the emission strengths between the biomass combustion and domestic coal combustion, the result would be more attractive. The authors only compared with the $SO_2$ emission factors between the biomass burning and coal cake combustion, however the emission factor of coal cake might be outdated, because raw bituminous is currently prevailing for cooking and heating in rural areas. The emission factors of various pollutants from combustion of raw bituminous in domestic stove have been reported (e.g. $SO_2$ emission factors of $4.16 \pm 1.36$ g $SO_2$ kg-1, Du, Q. et al. (2016), An important missing source of atmospheric carbonyl sulfide: Domestic coal combustion, Geophys. Res. Lett., 43(16), 8720–8727, doi:10.1002/2016GL070075; NMHCs (57 species) average emission factor of 2981.1 mg kg-1, Liu et al.(2017), Emission of volatile organic compounds from domestic coal stove with the actual alternation of flaming and smoldering combustion processes, Environmental Pollution 221, 385-391). Although the emission factors of $SO_2$ from the burning of corn and wheat straw is about 3-6 times less than that of coal combustion and of NMHCs is comparable to each other, the emissions of these pollutants from the biomass burning might largely exceed those from domestic coal combustion because the amount of the biomass burning might be one magnitude greater than that of domestic coal consumption in China. Therefore, greater attention should be paid on the emission of biomass burning for improving the air quality in China. The concentration of OH radical indirectly obtained by tracing the first order decay rate of toluene should represent its average concentration during the whole irradiation, why did you use the OH exposure of $(1.87\text{-}4.97) \times 10^{10}$ molecule cm-3s? Are your sure the lifetime of OH radical in the chamber is only 1s? I suggested to use the unit of average concentration $(1.87\text{-}4.97) \times 10^{10}$ molecule cm-3. Although the contribution of the 20 NMOGs to the SOA only accounted for 5-27.3% of the observed SOA mass, the increase of the SOA mass might not solely be ascribed to the aqueous-phase oxidation of alkenes, because the oxidation of the POM with more oxygen can also make evident contribution.

Page 16, line 366-368, The two OA enhancement ratios reported were evidently less

than those determined in this study, why did you concluded that the OA enhancement ratios determined were higher than those (0.7-2.9) for the combustion of vegetation, and comparable to those (0.7-6.9) for wood burning?

---

## Referee Comment (RC2) · Anonymous Referee #2 · 8 Sep 2017

Open burning of agricultural residues is a large source of both primary and secondary air pollutants in Asia and in China. Although many studies have been carried out, emission factors reported in previous study vary substantially due to differences in fuel types and combustion conditions. In addition, few studies have been performed to investigate the SOA formation from agricultural residues. To better understanding the effects of biomass burning on both primary and secondary pollution, this study comprehensively characterizes the primary emissions and determines SOA formation potential of emissions from the major agricultural residues in China, including corn, rice and wheat straws. Results from study would significantly improve our understanding the effects of agricultural residues on air quality. In addition, the results from this

study also provide constraints on estimation of the contributions of other sources to air pollution. Publication is recommended after the following comments and concerns are addressed.

General comments: Discussion is needed about why only a small fraction of observed SOA in this study was explained by the same set of speciated NMHCs which explain the majority of SOA in Bruns et al. (2016).

If the unexplained SOA is due to additional precursors, not quantified in this study, do these additional precursors substantially contribute to ozone formation? Based on the mass enhancement factor of 2.4-76 and the fact that similar emission factors for both measured NMHCs and PM, the amount of unmeasured NMHCs could be dramatically larger than the measured ones. Is there any study measuring both total NMHCs and speciated NMHCs from biomass burning? The difference between total NMHCs and speciated NMHCs is a useful indicator of additional precursors.

This study has covered a wide range of measurements and compared with measurements in past studies. However, what we can learn from this study, other than emissions factors and OA enhancement factors, is not clearly stated. In other works, what makes this paper significant is not clearly stated.

Specific comments: Line 136: define "purified dry air" Line 141-142: How was the water content determined? Line 152: change "diluted" to "dilute" Line 155: change "correct" to "determine" Line 161: The section of "Instrumentation" is actually "Characterization of primary emissions and secondary organic aerosol". In this section, I'd like to separate the description of the analysis of VOCs from other gases. Line 188: change "this instrument alternated" to "the HR-tof-AMS was operated by alternating" Line 188: change "one" to "other". Line 195: What is the AMS CE? Line 217: Is the denominator of the equation (2) the same as the numerator? Why do you need two equations to calculate this fuel based emission factor? Line 232: "NMHCs" should be "speciated NMHCs". In this study, the total NMHCs were not determined. Only a portion of them was speciated. Line 243: I suppose that the particle size evolves through the course of photo-oxidation experiments. Discussion is needed about whether the particle loss during the experiments can be corrected for using post measurements. Line 255: change "identified" to "quantified" Line 283: NMHCs were measured by two instruments: PTR-MS and GC-MS. Efforts are need to make sure readers can tell these measurements and follow the discussion. Line 297: Not all organic vapors were measured in this study. Do authors have an estimate of the unmeasured vapors across the three fuels and their ozone formation potential?

---

## Author Comment (AC1) · 27 Sep 2017

General comments:

This manuscript investigated the emissions of various primary pollutants and photochemical evolution from burning three types of agricultural residues (corn, rice and wheat straws) by using a 30m$^3$-smog chamber. The experimental design is reasonable, the results are reliable, and the conclusions are convincing. Considering the rare information on primary emissions and photochemical evolution of agricultural residues burning, the original data presented in this manuscript are very important for comprehensibly understanding the impact of the burning on the air quality, especially in China. The manuscript is well organized, and hence this reviewer recommends the manuscript be published in the journal.

**Reply**: Thanks for the positive comments. We have revised our manuscript after carefully reading the following constructive suggestions.

Specific comments:

[1] Both biomass burning and domestic coal combustion have been recognized to make evident contribution to deteriorating regional air quality especially in North China. If the authors had compared with the emission strengths between the biomass combustion and domestic coal combustion, the result would be more attractive. The authors only compared with the SO$_2$ emission factors between the biomass burning and coal cake combustion, however the emission factor of coal cake might be outdated, because raw bituminous is currently prevailing for cooking and heating in rural areas. The emission factors of various pollutants from combustion of raw bituminous in domestic stove have been reported (e.g. SO$_2$ emission factors of 4.16-1.36 g SO$_2$ kg$^{-1}$, Du, Q. et al. (2016), An important missing source of atmospheric carbonyl sulfide: Domestic coal combustion, Geophys. Res. Lett., 43(16), 8720–8727, doi:10.1002/2016GL070075; NMHCs (57 species) average emission factor of 2981.1 mg kg$^{-1}$, Liu et al.(2017), Emission of volatile organic compounds from domestic coal stove with the actual alternation of flaming and smoldering combustion processes, Environmental Pollution

221, 385-391). Although the emission factors of $SO_2$ from the burning of corn and wheat straw is about 3-6 times less than that of coal combustion and of NMHCs is comparable to each other, the emissions of these pollutants from the biomass burning might largely exceed those from domestic coal combustion because the amount of the biomass burning might be one magnitude greater than that of domestic coal consumption in China. Therefore, greater attention should be paid on the emission of biomass burning for improving the air quality in China.

**Reply**: Thanks. We agree that the comparison of primary emissions between agricultural residues burning and domestic coal combustion would help policy makers for the control of air pollution. As suggested, we have updated emission factors for coal combustion in the latest literatures, and added the following words in the revised "Conclusions" part:

"Both agricultural residues burning and domestic coal combustion have been recognized to contribute substantially to the deteriorating regional air quality especially in rural areas of China (Pan et al., 2015; Liu et al., 2016; Zhu et al., 2016). The emission factors of the speciated NMHCs, PM and $SO_2$ from combustion of raw bituminous, which is currently prevailing for cooking and heating in rural areas, have been reported to be 0.56-5.40, 25.49±2.30 and 2.43-5.36 g $kg^{-1}$, respectively (Du et al., 2016; Li et al., 2016; Liu et al., 2017). Annually burned crop residues and domestic coals were estimated to be 160 Tg (Li et al., 2016) and 99.6 Tg (NBSPRC, 2014) in China. Therefore, with the emission factors of the speciated NMHCs (2.47-5.04 g $kg^{-1}$), PM (3.73-6.36 g $kg^{-1}$) and $SO_2$ (0.07-0.99 g $kg^{-1}$) measured for agricultural residues burning in this study, agricultural residues burning might emit more NMHCs, but less primary PM and $SO_2$ than domestic coal burning on a national scale."

[2] The concentration of OH radical indirectly obtained by tracing the first order decay rate of toluene should represent its average concentration during the whole irradiation, why did you use the OH exposure of (1.87-4.97)×$10^{10}$ molecule $cm^{-3}$ s? Are your sure

the lifetime of OH radical in the chamber is only 1s? I suggested to use the unit of average concentration $(1.87-4.97) \times 10^{10}$ molecule $cm^{-3}$.

**Reply**: In fact, the average OH concentration in this manuscript was calculated every 2 minutes by continuous monitoring of toluene concentration through PTR-TOF-MS, thus the average "OH exposure" every 2 min was calculated as the product of average OH concentration and the time interval. OH exposure, indicating the atmospheric oxidation power that a pollutant undergoes, is a parameter that has been widely used in the chamber studies (e.g., Hennigan et al., 2010; Tiitta et al., 2016; Tkacik et al., 2017). In the revised manuscript, we have revised the manuscript to include this detailed information about how the average OH concentration was calculated: "The decay of toluene measured by PTR-TOF-MS was used to derive the average OH radical concentrations for every 2 min during each experiment, and the integrated OH exposure was calculated as the product of the OH concentration and the time interval."

[3] Although the contribution of the 20 NMOGs to the SOA only accounted for 5-27.3% of the observed SOA mass, the increase of the SOA mass might not solely be ascribed to the aqueous-phase oxidation of alkenes, because the oxidation of the POM with more oxygen can also make evident contribution.

**Reply**: Thanks for the comments. The possible contribution from the oxidation of POM has been included in the revised manuscript, and some other reasons for the discrepancy have also been added:

"It is noted that although over 80 VOCs species were quantified by the GC-MSD/FID and the PTR-TOF-MS in this study, only 20 species among them were taken into the SOA prediction because of the lack of published data for SOA yields. The unaccounted VOC species might be a reason for the discrepancy. On the other hand, as indicated by Deng et al. (2017), SOA yields obtained from chamber studies in purified air matrix might be lower than that in real ambient air matrix. Consequently, using SOA yields from studies in purified air matrix might also under predict SOA yields in the complex

biomass burning plume matrix. Moreover, oxidation of particulate organic matters (POM), like semi-volatile organic compounds (SVOC) and intermediate volatility organic compounds (IVOC), would also contribute substantially to SOA formation (Presto et al., 2009; Zhao et al., 2014), yet this is not accounted for in our prediction..”

[4] (Page 16, line 366-368) The two OA enhancement ratios reported were evidently less than those determined in this study, why did you concluded that the OA enhancement ratios determined were higher than those (0.7-2.9) for the combustion of vegetation, and comparable to those (0.7-6.9) for wood burning?

**Reply**: The OA enhancement ratio in our study should be 2.4-7.6 rather than 2.4-76 in this sentence. We are quite sorry for this typo and have corrected it in the revised manuscript.

**References**

[revised manuscript text omitted]

---

## Author Comment (AC2) · 27 Sep 2017

General comments:

[1] Open burning of agricultural residues is a large source of both primary and secondary air pollutants in Asia and in China. Although many studies have been carried out, emission factors reported in previous study vary substantially due to differences in fuel types and combustion conditions. In addition, few studies have been performed to investigate the SOA formation from agricultural residues. To better understanding the effects of biomass burning on both primary and secondary pollution, this study comprehensively characterizes the primary emissions and determines SOA formation potential of emissions from the major agricultural residues in China, including corn, rice and wheat straws. Results from study would significantly improve our understanding the effects of agricultural residues on air quality. In addition, the results from this study also provide constraints on estimation of the contributions of other sources to air pollution. Publication is recommended after the following comments and concerns are addressed.

**Reply**: Thanks for the comments. We have revised our manuscript with your constructive comments and suggestions below.

[2] Discussion is needed about why only a small fraction of observed SOA in this study was explained by the same set of speciated NMHCs which explain the majority of SOA in Bruns et al. (2016).

**Reply**: Only a small fraction of observed SOA in this study was explained by the same set of speciated NMHCs, which explained the majority of SOA in Bruns et al. (2016). Recent studies indicated that IVOC and SVOC may contribute substantially to SOA. There might be at least SVOC in the biomass burning plumes. As indicated by the AMS data, CH-family or hydrocarbon was the major component of POA in the initial biomass burning plume (Figure 6a), after photo-oxidation they decreased in aged OA. This family of SVOC could be oxidized to form SOA. Considering the probable contribution from IVOC/SVOC to SOA formation, it would be reasonable that we observed the

discrepancy. In the revised manuscript, we have added an explanation as below:

"It is noted that although over 80 VOCs species were quantified by the GC-MSD/FID and the PTR-TOF-MS in this study, only 20 species among them were taken into the SOA prediction because of the lack of published data for SOA yields. The unaccounted VOC species might be a reason for the discrepancy. On the other hand, as indicated by Deng et al. (2017), SOA yields obtained from chamber studies in purified air matrix might be lower than that in real ambient air matrix. Consequently, using SOA yields from studies in purified air matrix might also under predict SOA yields in the complex biomass burning plume matrix. Moreover, oxidation of particulate organic matters (POM), like semi-volatile organic compounds (SVOC) and intermediate volatility organic compounds (IVOC), would also contribute substantially to SOA formation (Presto et al., 2009; Zhao et al., 2014), yet this is not accounted for in our prediction."

[3] If the unexplained SOA is due to additional precursors, not quantified in this study, do these additional precursors substantially contribute to ozone formation? Based on the mass enhancement factor of 2.4-76 and the fact that similar emission factors for both measured NMHCs and PM, the amount of unmeasured NMHCs could be dramatically larger than the measured ones. Is there any study measuring both total NMHCs and speciated NMHCs from biomass burning? The difference between total NMHCs and speciated NMHCs is a useful indicator of additional precursors.

**Reply**: To our best knowledge, emissions of total NMHCs for agricultural residues burning has not yet been reported, and the amount of NMHCs quantified in this study (67 species) outnumbered those in other studies, such as 52 species measured by Li et al. (2009) and 56 species measured by Wang et al. (2014). We fully agree that measuring total NMHCs would help explaining the data. This is a very good suggestion and we are thinking how we can get it done in our future study. We think the additional precursors would also contribute substantially to ozone formation. A considerable amount of NMOG species, such like OVOCs, were detected by PTR-TOF-MS but not

reported in this manuscript. Unmeasured NMHCs would also be oxidized to produce OVOCs, which may contribute substantially to ozone formation. For example, formaldehyde and acetaldehyde were among the most abundant NMOGs species detected by PTR-TOF-MS in the initial biomass burning plumes, and they also have relatively high maximum incremental reactivity (MIR) values (Carter, 2008). We have added the following statement in the revised manuscript:

"It is noted that oxygen-containing organic vapors in agricultural residues burning plumes could also have large ozone formation potentials. For example, the OFPs of formaldehyde and acetaldehyde for all experiments were 0.57-2.46 times of the 67 speciated NMHCs."

[4] This study has covered a wide range of measurements and compared with measurements in past studies. However, what we can learn from this study, other than emissions factors and OA enhancement factors, is not clearly stated. In other works, what makes this paper significant is not clearly stated.

**Reply**: We think that this paper gives something new in three aspects:

(1) The emission factors were measured at ambient-level dilution ratios. Thus the errors caused by the evaporation of SVOCs from the particle phase to the gas phase were avoided.

(2) More than 60% of SOA mass cannot be explained by the known precursors.

(3) The $f_{60}$ values in AMS spectrum of primary agricultural residues emissions were lower than those from field campaigns. The $f_{60}$ value is often regarded as the biomass burning marker, so the present constraints on it need to be reconsidered.

Besides, as suggested by the anonymous referee #1, a comparison between agricultural residues burning and domestic coal burning in China were done to help policy makers in air pollution control especially in rural areas.

Specific comments:

[1] Line 136: define "purified dry air"

**Reply**: The following sentences have been added to the revised manuscript: "Compressed indoor air is forced through an air dryer (FXe1; Atlas Copco; Sweden) and a series of bed scrubbers containing activated carbon, Purafil, Hopcalite and allochroic silica gel, followed by a PTFE filter to provide the source of purified air with a flow rate of 100 L min$^{-1}$. The purified dry air contains <1 ppb $NO_x$, $O_3$ and carbonyl compounds, <5 ppb NMHCs and no detectable particles with relative humidity <5%."

[2] Line 141-142: How was the water content determined?

**Reply**: The following sentences have been added to the revised manuscript: "The water content of crop residues was measured by using the method recommended by Liao et al. (2004). The weight of straws were weighed before and after baking in a stove at 105°C for 24 h, and the difference in weights was calculated to be the weight of the water in the crop residues. Water content was the quotient of the water weight and the whole weight of the straws."

[3] Line 152: change "diluted" to "dilute"

Line 155: change "correct" to "determine"

Line 188: change "this instrument alternated"± to "the HR-TOF-AMS was operated by alternating"

Line 188: change "one" to "other"

Line 255: change "identified" to "quantified"

**Reply**: Thank you for the suggestions. We have corrected those errors in the revised manuscript.

[4] Line 161: The section of "Instrumentation" is actually "Characterization of primary emissions and secondary organic aerosol". In this section, I'd like to separate the description of the analysis of VOCs from other gases.

**Reply**: Thank you for the suggestion. We have separated the description of the analysis of VOCs from other gases. The first paragraph of section 2.2 has been corrected as follows:

"Commercial instruments were used for online monitoring of $NO_x$ (EC9841T, Ecotech, Australia), $NH_3$ (Model 911-0016, Los Gatos Research, USA) and $SO_2$ (Model 43i, Thermo Scientific, USA). $CH_4$ and CO were analyzed offline using a gas chromatography (Agilent 6980GC, USA) coupled with a flame ionization detector and a packed column (5A molecular sieve 60/80 mesh, 3 m × 1/8 in) (Zhang et al., 2012), and $CO_2$ was analyzed using a HP 4890D gas chromatograph (Yi et al., 2007). The detection limits were all less than 30 ppbv for $CH_4$, CO and $CO_2$. The relative standard deviations (RSDs) of CO and $CO_2$ measurements were both less than 3% based on seven duplicate injection of 1.0 ppmv standards (Spectra Gases Inc, USA).

Volatile organic compounds (VOCs) were continuously measured using a proton-transfer-reaction time-of-flight mass spectrometer (PTR-TOF-MS; Model 2000, Ionicon Analytik GmbH, Austria). Calibration of the PTR-TOF-MS was performed every few weeks using a certified custom-made standard mixture of VOCs (Ionicon Analytik Gmbh, Austria) that were dynamically diluted to 6 levels (2, 5, 10, 20, 50 and 100 ppbv). Methanol, acetonitrile, acetaldehyde, acrolein, acetone, isoprene, crotonaldehyde, 2-butanone, benzene, toluene, o-xylene, chlorobenzene and α-pinene were included in the calibration mixture. Their sensitivities, indicated by the ratio of the normalized counts per second to the concentration levels of the VOCs in ppbv, were used to convert the raw PTR-TOF-MS signal to concentration (Huang et al., 2016). Quantification of the compounds that were not included in the mixture was performed by using calculated mass-dependent sensitivities based on the measured sensitivities (Stockwell et al., 2015). Mass-dependent sensitivities were linearly fitted for oxygen-containing compounds and the remaining compounds separately. The decay of toluene measured by PTR-TOF-MS was used to derive the OH radical concentrations for every 2 min during each experiment, and the OH exposure was calculated as the product of

the OH concentration and the time interval. Continuous monitoring of 20 SOA precursors (including 9 NMHCs and 11 oxygen-containing VOCs) from PTR-TOF-MS provided us with data to do the SOA prediction discussed in the Sect 2.3.5 and 3.3.2. Air samples were also collected from the chamber reactor using 2-Liter electro-polished stainless-steel canisters before and after smoke injection. In total 67 $C_2$-$C_{12}$ NMHCs were measured (Table S1) using an Agilent 5973N gas chromatography mass-selective detector/flame ionization detector (GC-MSD/FID; Agilent Technologies, USA) coupled to a Preconcentrator (Model 7100, Entech Instruments Inc., USA), and analytical procedures have been detailed elsewhere (Wang and Wu, 2008; Zhang et al., 2010; Zhang et al., 2012). Results from GC-MSD/FID were used to quantify the emission factors of 67 NMHCs discussed in the Sect 3.1."

[5] Line 195: What is the AMS CE?

**Reply**: AMS tends to underestimate the PM mass due to the transmission efficiency (Liu et al., 2007) and the AMS collection efficiency (Gordon et al., 2014). Besides, the black carbon, which is an important part of biomass burning particles, can hardly be captured because it doesn't rapidly vaporize in the vaporizer of AMS. These factors would lead to the discrepancy between the AMS data and SMPS data. AMS collection efficiency (CE) is calculated as the quotient of the total mass measure by AMS and the mass difference between the SMPS and the aethalometer. By dividing the AMS CE, the AMS data were corrected. Experiment-specific CEs ranged from 0.20 to 0.41 in this study.

[6] Line 217: Is the denominator of the equation (2) the same as the numerator? Why do you need two equations to calculate this fuel based emission factor?

**Reply**: In fact, the carbon mass after burning will be distributed in both ash and in the gas phase, the equation (3) defines how to calculate the emission factor of the total carbon mass in the gas phase (EF$_c$) by elemental and gravitational analysis. The

equation (3) was often ignored in previous papers because the ash part was neglected while we want to clarify it explicitly here. In the right side of equation (2), the part $\frac{m_i}{\Delta[CO_2]+\Delta[CO]+\Delta[PM_C]+\Delta[HC]}$ means the mass ratio of the $i$th species and the measured total carbon mass, and the product of this ratio and $EF_c$ was defined as the emission factor of the $i$th species.

[7] Line 232: "NMHCs" should be "speciated NMHCs". In this study, the total NMHCs were not determined. Only a portion of them was speciated.

**Reply**: Thanks for your suggestion. We have changed "NMHCs" to "speciated NMHCs" in the whole manuscript.

[8] Line 243: I suppose that the particle size evolves through the course of photo-oxidation experiments. Discussion is needed about whether the particle loss during the experiments can be corrected for using post measurements.

**Reply**: We agree that the particle wall loss rate is size-dependent. The relationship between the particle loss rate and the diameter was shown in Figure 1. From burn 1-6, the uncorrected particle size grew from 68 to 92, 71 to 148, 57 to 91, 61 to 98, 82 to 150 and 57 to 105 nm during the photo-oxidation. Assuming that the wall loss rates during the whole photoreaction should correspond to the averaged size of the primary particles and the aged particles, they were underestimated by 5.4%, -2.0%, 8.8%, 7.3%, -2.8% and 7.8%, respectively, when we use the wall loss rates after lights were off for simplification. So it might be acceptable to use post measurements to determine the particle wall loss rate, though an error range of ~±9% should be noted. The discussion above have been added to the supplement material of this manuscript.

[Figure]

**Fig 1.** The relationship between wall loss rate and the particle diameter. The data were calculated from burn 6 in which wheat straws were burned. The fitting was based on the equation suggested by Takekawa et al. (2003).

[9] Line 283: NMHCs were measured by two instruments: PTR-MS and GC-MS. Efforts are need to make sure readers can tell these measurements and follow the discussion.

**Reply**: We have added the following statement to Sect 2.2 where PTR-TOF-MS is introduced: "Continuous monitoring of 20 SOA precursors (including 9 NMHCs and 11 oxygen-containing VOCs) from PTR-TOF-MS provided us with data to do the SOA prediction discussed in the Section 2.3.5 and 3.3.2." Also in Sect 2.2 where GC-MSD/FID is introduced, the following sentence has been added: "Results from GC-MSD/FID were used to quantify the emission factors of 67 NMHCs discussed in the Section 3.1."

[10] Line 297: Not all organic vapors were measured in this study. Do authors have an estimate of the unmeasured vapors across the three fuels and their ozone formation potential?

**Reply**: As replying your comment above, some species of NMOGs detected by PTR-TOF-MS are not included in this manuscript for SOA prediction, but they did have both high EFs and high ozone formation potentials. But even with the help of GC-MSD/FID and PTR-TOF-MS, we are not sure to be able to identify and precisely quantify all organic vapors generated from agricultural residues burning because of difficulties in separating isomers from each other (Hatch et al., 2017; Bruns et al., 2017) and because of some intermediate volatility organic compounds are not detected but may also contribute to form SOA and ozone. In fact we are also very interested in this topic and what to know if the "traditional" ozone precursors could explain the ozone formation during our photo-oxidation experiments, and for this we may need to use the MCM model. We have started to write a paper on this aspect. In this manuscript we just put our focus on that if the known precursors could explain the SOA formed during photo-oxidation.

**References**

Bruns, E. A., El Haddad, I., Slowik, J. G., Kilic, D., Klein, F., Baltensperger, U., and Prevot, A. S. H.: Identification of significant precursor gases of secondary organic aerosols from residential wood combustion, Sci. Rep., 6, doi:10.1038/srep27881, 2016.

Bruns, E. A., Slowik, J. G., El Haddad, I., Kilic, D., Klein, F., Dommen, J., Temime-Roussel, B., Marchand, N., Baltensperger, U., and Prevot, A. S. H.: Characterization of gas-phase organics using proton transfer reaction time-of-flight mass spectrometry: fresh and aged residential wood combustion emissions, Atmos. Chem. Phys., 17, 705-720, doi:10.5194/acp-17-705-2017, 2017.

Carter, W. P. L.: Reactivity estimates for selected consumer product compounds, Air

resources Board, California, Contract No. 06-408, 72-99, 2008.

Deng, W., Liu, T., Zhang, Y., Situ, S., Hu, Q., He, Q., Zhang, Z., Lü, S., Bi, X., Wang, X., Boreave, A., George, C., Ding, X., and Wang, X.: Secondary organic aerosol formation from photo-oxidation of toluene with $NO_x$ and $SO_2$: Chamber simulation with purified air versus urban ambient air as matrix, Atmos. Environ., 150, 67-76, doi:10.1016/j.atmosenv.2016.11.047, 2017.

Hatch, L. E., Yokelson, R. J., Stockwell, C. E., Veres, P. R., Simpson, I. J., Blake, D. R., Orlando, J. J., and Barsanti, K. C.: Multi-instrument comparison and compilation of non-methane organic gas emissions from biomass burning and implications for smoke-derived secondary organic aerosol precursors, Atmos. Chem. Phys., 17, 1471-1489, doi:10.5194/acp-17-1471-2017, 2017.

Gordon, T. D., Presto, A. A., May, A. A., Nguyen, N. T., Lipsky, E. M., Donahue, N. M., Gutierrez, A., Zhang, M., Maddox, C., Rieger, P., Chattopadhyay, S., Maldonado, H., Maricq, M. M., and Robinson, A. L.: Secondary organic aerosol formation exceeds primary particulate matter emissions for light-duty gasoline vehicles, Atmos. Chem. Phys., 14, 4661-4678, doi:10.5194/acp-14-4661-2014, 2014.

Li, X. H., Wang, S. X., Duan, L., and Hao, J. M.: Characterization of non-methane hydrocarbons emitted from open burning of wheat straw and corn stover in China, Environ. Res. Lett., 4, 7, doi:10.1088/1748-9326/4/4/044015, 2009.

Liao, C. P., Wu, C. Z., Yanyongjie, and Huang, H. T.: Chemical elemental characteristics of biomass fuels in China, Biomass Bioenergy, 27, 119-130, doi:f10.1016/j.biombioe.2004.01.002, 2004.

Liu, P. S. K., Deng, R., Smith, K. A., Williams, L. R., Jayne, J. T., Canagaratna, M. R., Moore, K., Onasch, T. B., Worsnop, D. R., and Deshler, T.: Transmission efficiency of an aerodynamic focusing lens system: Comparison of model calculations and laboratory measurements for the Aerodyne Aerosol Mass Spectrometer, Aerosol Sci. Technol., 41, 721-733,

doi:10.1080/02786820701422278, 2007.

Presto, A. A., Miracolo, M. A., Kroll, J. H., Worsnop, D. R., Robinson, A. L., and Donahue, N. M.: Intermediate-volatility organic compounds: A potential source of ambient oxidized organic aerosol, Environ. Sci. Technol., 43, 4744-4749, doi:10.1021/es803219q, 2009.

Takekawa, H., Minoura, H., and Yamazaki, S.: Temperature dependence of secondary organic aerosol formation by photo-oxidation of hydrocarbons, Atmos. Environ., 37, 3413-3424, doi:10.1016/s1352-2310(03)00359-5, 2003.

Wang, H., Lou, S., Huang, C., Qiao, L., Tang, X., Chen, C., Zeng, L., Wang, Q., Zhou, M., Lu, S., and Yu, X.: Source profiles of volatile organic compounds from biomass burning in Yangtze River Delta, China, Aerosol Air Qual. Res., 14, 818-828, doi:10.4209/aaqr.2013.05.0174, 2014.

Zhao, Y. L., Hennigan, C. J., May, A. A., Tkacik, D. S., de Gouw, J. A., Gilman, J. B., Kuster, W. C., Borbon, A., and Robinson, A. L.: Intermediate-volatility organic compounds: A large source of secondary organic aerosol, Environ. Sci. Technol., 48, 13743-13750, doi:10.1021/es5035188, 2014.